# Life Satisfaction and Self-Esteem in Older Adults Engaging in Formal Volunteering: A Cross-Sectional Study in Taiwan

**DOI:** 10.3390/ijerph20064934

**Published:** 2023-03-10

**Authors:** Jo-Tzu Chu, Malcolm Koo

**Affiliations:** 1Graduate Institute of Long-Term Care, Tzu Chi University of Science and Technology, Hualien City 970302, Taiwan; 2Dalla Lana School of Public Health, University of Toronto, Toronto, ON M5T 3M7, Canada

**Keywords:** mental health, life satisfaction, well-being, social participation, volunteering, older adults

## Abstract

Previous research has reported an association between life satisfaction, self-esteem, and volunteering. However, it is unclear whether self-esteem is associated with life satisfaction in older adults who are already engaged in volunteering. Therefore, the present study aimed to investigate the association between life satisfaction and self-esteem in older adults who were formally volunteering at a non-governmental organization in Taiwan. A cross-sectional study was conducted on 186 formal volunteers aged ≥ 65 years who were recruited from the Keelung chapter of the Buddhist Compassion Relief Tzu Chi Foundation in Taiwan. A hierarchical stepwise linear regression was used to examine the association between scores on the Satisfaction With Life Scale (SWLS) with the Rosenberg Self-Esteem Scale (RSES) and the Hedonic and Eudaimonic Motives for Activities-Revised (HEMA-R) scale. The results showed that SWLS was significantly associated with RSES score (standardized beta (std. β) = 0.199, *p* = 0.003), the eudaimonic subscale score of the HEMA-R (std. β = 0.353, *p* < 0.001), a vegetarian diet (std. β = 0.143, *p* = 0.027), and volunteering for five days or more a week (std. β = 0.161, *p* = 0.011). In conclusion, improving self-esteem and promoting eudaimonic motives in older adults who are formally volunteering could be effective strategies for enhancing their levels of life satisfaction.

## 1. Introduction

Population aging is a prevalent global phenomenon. It has been projected that the proportion of the global population aged 65 and above will increase from 10% in 2022 to 16% in 2050 [1]. The United Nations Programme on Ageing has identified advancing health and well-being in old age as a key priority in its Research Agenda on Ageing for the 21st Century [2]. Consequently, identifying predictors of well-being among older adults is a crucial step in promoting successful aging in this population [3].

Well-being is a complex construct that encompassing various psychological and social dimensions [4]. It can be considered a general term that encompasses all aspects of human life domains that contribute to what is known as a “good life”, including physical, mental, and social aspects, as defined by the World Health Organization [5]. Subjective well-being, as defined by Diener, refers to people’s evaluation of their own lives regarding their cognitions and feelings [6]. Research suggested that subjective well-being was positively associated with physical health and longevity in healthy populations [7,8]. In the present study, life satisfaction, which is the cognitive component of subjective well-being, was used to assess the subjective evaluation of an individual’s overall quality of life [9].

Self-esteem, which is the positive or negative attitude an individual holds towards themselves [10], is a strong predictor of subjective well-being [11,12] and life satisfaction [9]. It can also be defined as the subjective assessment of one’s own worth [13]. Individuals with higher levels of self-esteem tend to have more positive relationships and higher levels of social support, which can contribute to higher levels of life satisfaction [10,14]. Conversely, low self-esteem is associated with depression [15] and non-suicidal self-injury [16], which can adversely impact life satisfaction. A meta-analysis of 50 articles involving 29,839 participants and a secondary analysis of 74,381 participants from four national studies showed that self-esteem stability was low during childhood, rose throughout adolescence and young adulthood, and declined during midlife and old age [17].

The positive relationship between self-esteem and life satisfaction can be explained by the Self-Evaluation Maintenance model, proposed by Abraham Tesser [18]. According to this model, the relationship between self-esteem and life satisfaction is driven by two key processes: the reflection process and the comparison process. Individuals with high self-esteem are more likely to reflect positively on themselves and less likely to engage in the comparison process, which contributes to their greater sense of life satisfaction. Conversely, individuals with low self-esteem are more likely to compare themselves to others who perform better than them and less likely to engage in the reflection process, which contributes to their lower levels of life satisfaction.

Volunteering has been associated with positive aging [19], physical health [20,21], and subjective well-being [22]. A study based on the World Values Survey dataset found that volunteering had a beneficial effect on the well-being of older volunteers in Hong Kong, Japan, Singapore, South Korea, and Taiwan [23]. In addition, evidence from a systematic review and meta-analysis of 40 studies suggested that volunteering could benefit mental health and survival. Nevertheless, the delivery of volunteering, such as frequency, dose, and type of activity, needs to be further established to yield optimal health benefits [24]. 

Hedonic and eudaimonic well-being are two distinct but related concepts that have been extensively studied in the field of positive psychology. However, previous research has reported inconsistent results regarding whether hedonic and eudaimonic orientations promote well-being [25,26,27]; the Hedonic and Eudaimonic Motives for Activities-Revised scale (HEMA-R) was used in this study to assess participants’ orientations towards well-being [28]. Eudaimonia, as defined by Huta, involves striving to develop the best in oneself in ways that are congruent with one’s values and true self. In contrast, hedonia involves striving to experience pleasure, enjoyment, and comfort [29]. As such, orientations to well-being can be classified into two categories: hedonic well-being, which involves feelings of joy, pleasure, and happiness and eudaimonic well-being, which encompasses a sense of purpose, meaning in life, and self-realization. By including both constructs as predictors in our analysis, we hope to gain a better understanding of the independent contributions of each construct to life satisfaction.

Although several studies have reported an association between life satisfaction and self-esteem in the context of volunteering, to our knowledge, it is not known whether self-esteem is associated with life satisfaction in older adults who were already engaged in volunteering. Therefore, to address this research gap, the present study aimed to explore the association between life satisfaction and self-esteem among older adults who were formally volunteering at a non-governmental organization in Taiwan. 

## 2. Materials and Methods

### 2.1. Study Design and Participants

A cross-sectional study design was employed in this study. Eligible participants were recruited using convenience sampling from the Keelung chapter of the Buddhist Compassion Relief Tzu Chi Foundation in Taiwan. To conduct the survey, a trained interviewer visited the chapter office two to three times per week between 1 and 30 November 2022. A total of 191 questionnaires was distributed, but five contained missing responses, resulting in 186 questionnaires being included for analysis.

Inclusion criteria were as follows: age 65 years and older, currently engaged in formal volunteering, able to communicate in Mandarin or Taiwanese, and able to complete the study questionnaires. Structured paper-based questionnaires were administered in person by the interviewer. Each respondent was offered a box of medical face masks valued at 100 New Taiwan Dollars (approximately USD 3) for participating in the study.

Sample size estimation was performed using G*Power 3.1.9.4 software (Heinrich Heine University Düsseldorf, Düsseldorf, Germany), which indicated that a sample size of 185 participants was required to detect a small–medium effect size (Cohen *f*^2^ = 0.085) in a multiple linear regression model with eight predictors, at an alpha level of 0.05 and a power of 80%. 

Prior to the study, ethical approval was sought from the Research Ethics Committee of Hualien Tzu Chi Hospital, Buddhist Tzu Chi Medical Foundation (protocol code: IRB111-190-B). 

### 2.2. Measurement

The independent variables in this study consisted of four parts: (1) participants’ basic characteristics (sex, age, educational level, marital status, employment status, dietary habits, chronic disease, and sleep duration); (2) volunteering information (duration of volunteering, frequency of volunteering, and the main type of volunteering; (3) the Hedonic and Eudaimonic Motives for Activities-Revised scale (HEMA-R) [28]; and (4) the Rosenberg Self-Esteem Scale [10].

The Rosenberg Self-Esteem Scale is a widely used measure of self-esteem that aims to predict or explain individual behavior based on self-worth, self-motivation, and confidence. The scale comprises ten items, five positively and five negatively worded, rated on a 4-point Likert response format, ranging from 1 (strongly disagree) to 4 (strongly agree). The total score ranges from 10 to 40, with higher scores indicating higher self-esteem [10]. The McDonald’s Ω coefficient of the Rosenberg Self-Esteem Scale in this study was 0.82 (95% CI 0.76, 0.86).

The Chinese version of the Hedonic and Eudaimonic Motives for Activities-Revised scale (HEMA-R) was used to measure the hedonic and eudaimonic orientations of the respondents when approaching their activities [30]. Respondents were asked to rate on a 7-point Likert-type scale from 1 (not at all) to 7 (very much) to what degree they typically approach their activities. The HEMA-R comprises ten items, five assessing eudaimonic motives (e.g., Seeking to use the best in yourself) and five assessing hedonic motives (e.g., Seeking enjoyment). The McDonald’s Ω coefficients of the hedonic subscale and eudaimonic subscale of the HEMA-R in this study were 0.69 (95% CI 0.60, 0.76) and 0.82 (95% CI 0.75, 0.87), respectively. 

The outcome variable in this study was the levels of life satisfaction measured by the widely used Satisfaction with Life Scale (SWLS) (http://labs.psychology.illinois.edu/~ediener/SWLS.html, accessed on 27 January 2023). In 1985, Diener et al. developed the SWLS, with five statements, to measure global cognitive judgments of satisfaction with one’s life [31]. The psychometric properties of the SWLS have been extensively studied in various populations [32,33,34,35,36,37,38]. The five items of the SWLS were rated using a 7-point Likert scale, ranging from 1 (strongly disagree) to 7 (strongly agree), with a total score ranging from 5 to 35. Pavot and Diener proposed that a score of 20 represents the neutral point on the scale, indicating that the respondent is equally satisfied and dissatisfied [38]. A cross-national analysis supported a single-factor structure and measurement invariance of the SWLS across the United States, England, and Japan [35]. Moreover, in a nationally representative sample of 4795 adults in China, the Cronbach’s α of the scale was found to be 0.88 [36]. A study on 476 undergraduate students in Taiwan revealed that the SWLS Taiwan version had the property of factorial invariance across sex, and the confirmatory factor analysis supported the single-factor structure [37]. The McDonald’s Ω of the SWLS in this study was 0.78 (95% CI 0.71, 0.82).

### 2.3. Data Analysis

Data were presented as the mean and standard deviation (SD) or as the number of participants and percentage (%), as appropriate. The hierarchical stepwise linear regression was used to examine the association between SWLS score and independent variables. Basic characteristics of the participants, including sex, age group, educational level, marital status, employment status, dietary habits, chronic disease, and sleep duration were first evaluated as control variables in Model 1. Subsequently, volunteering-related variables, including duration, frequency, and main type of volunteering, were evaluated as the second block (Model 2). Third, the Rosenberg Self-Esteem Scale score was introduced as the third block (Model 3). Finally, the hedonic and eudaimonic subscales of the HEMA-R were evaluated as the fourth block (Model 4). 

The variance inflation factor (VIF) was used to evaluate multicollinearity among the independent variables in the final model. The Durbin–Watson statistic was used to assess autocorrelation in residuals. In addition, the internal consistency reliability of the scales was assessed using McDonald’s omega (Ω) coefficient, and a 95% confidence interval (CI) was established based on 1000 bootstrap samples. All statistical analyses were performed using IBM SPSS Statistics, version 28.0.1.1 (IBM Corporation, Armonk, NY, USA) with statistical significance set at *p* < 0.05.

## 3. Results

### 3.1. Demographic Characteristics

The mean age of the 186 respondents was 72.3 years (SD 4.8 years), with 21.5% male. More than half of the sample (57.5%) had attained a high school education or higher, 64.5% were married, 12.4% were employed full-time or part-time, 67.7% of respondents adhered to a strict or lacto-ovo-vegetarian diet, 46.2% had at least one chronic disease, and 64.5% reported sleeping less than seven hours per night. In terms of volunteering, 45.7% had volunteered for more than 20 years and 22.6% were volunteering for five or more days per week. The main type of volunteering was environmental protection (46.8%), followed by charity work (16.1%) and community work (12.9%) (Table 1).

### 3.2. Descriptive Statistics for the Scores of the Rosenberg Self-Esteem Scale, Hedonic and Eudaimonic Motives for Activities-Revised Scale, and Satisfaction with Life Scale

Table 2 summarizes the descriptive statistics for the scores of Rosenberg Self-Esteem Scale, the two subscales of the HEMA-R, and Satisfaction with Life Scale. The mean scores were 29.5, 23.3, 27.9, and 26.8, respectively.

Table 3 shows the results of the hierarchical linear stepwise regression analysis of the SWLS score. Model 1 included the basic characteristics of the respondents, and only adherence to a vegetarian diet (*p* < 0.001) remained in the model, explaining 6.3% of the variance in the SWLS score. Model 2 added three volunteering-related variables, and only the frequency of volunteering for five or more days per week (*p* = 0.002) was significantly associated with the SWLS score. The addition of these variables significantly increased the adjusted R^2^ to 0.104. In Model 3, the two subscales of the HEMA-R were added to the model, and only the score of the eudaimonic subscale (*p* < 0.001) was significantly associated with the SWLS score. The inclusion of this variable further increased the R^2^ to 0.263. Finally, the Rosenberg Self-Esteem Scale score was added to the model, and it was significantly associated with the SWLS score (*p* = 0.003). The adjusted R^2^ of the final model was 0.294. The Durbin–Watson statistic was 2.03, indicating no autocorrelation was present. Moreover, the VIF values for the five variables in the final model ranged from 1.03 to 1.19, indicating the absence of multicollinearity between the independent variables.

## 4. Discussion

This cross-sectional study investigated the association between self-esteem and life satisfaction in older adults who were formally volunteering in a non-governmental organization in Taiwan. Our results showed that higher self-esteem was significantly associated with high levels of life satisfaction, as measured by the SWLS, after adjusting for other covariates, including a vegetarian dietary habit, frequency of volunteering, and the eudaimonic motivations for approach activities. Previous research has shown that self-esteem was a strong predictor of life satisfaction in youth people [39] and adults [11]. However, fewer studies have investigated the relationship in older adults. A sub-study of the European Study of Adult Well-Being project, including six European countries on 2195 older adults, revealed that high self-esteem, measured by the Rosenberg’s Self-Esteem Scale, was significantly associated with high life satisfaction [40]. As volunteering is associated with improved life satisfaction [41], our study further showed that self-esteem is positively related to life satisfaction among older adults who were already engaged in volunteering. Our findings are also consistent with those of a survey of 366 volunteers of St. John Ambulance Malaysia, which found that the self-esteem of volunteers was significantly related to life satisfaction [42]. 

The association between self-esteem and life satisfaction observed in our participants could be explained by the Self-Evaluation Maintenance model [18]. Volunteer work can be a source of positive experiences, such as feelings of accomplishment, social connection, and a sense of purpose. According to the Self-Evaluation Maintenance model, these positive experiences could contribute to higher levels of self-esteem, as individuals maintain positive self-evaluations based on their perceived competence within the volunteer community. In addition, individuals may compare themselves to others who are perceived as less competent or less engaged in volunteer work, which will boost their own self-evaluations. This social comparison process could contribute to higher levels of self-esteem, and, in turn, higher levels of life satisfaction. Therefore, future studies could explore approaches to enhance self-esteem among older volunteers, and, thereby, improve their life satisfaction levels. For instance, group integrative reminiscence therapy was found to have a positive impact on self-esteem, life satisfaction, and depressive symptoms in institutionalized older veterans [43]. Moreover, an 8-week life review group program was shown to significantly improve self-esteem and life satisfaction in a randomized controlled trial of older men from a Veteran’s home in Taiwan, compared to the control group [44].

The present study also showed that engaging in formal volunteering for five or more days per week was significantly associated with higher levels of life satisfaction among older adults. This finding is consistent with a previous study based on the Americans’ Changing Lives survey, which found that increasing time commitment to productive activities, including volunteering, was significantly associated with higher levels of life satisfaction in respondents aged 60 years and older [45]. Another study that investigated five Asian societies with aging populations also concluded that older people should be encouraged to engage in longer hours of voluntary services. Merely joining volunteer organizations without a strong commitment did not appear to enhance well-being [23]. While the psychological beneficial effects of volunteering on well-being have been extensively documented [46,47], the present study revealed that for those who were already volunteering, engaging in formal volunteering five or more days per week, contributed to greater life satisfaction compared to those engaged in less than five days per week.

One novel finding from our study is that vegetarianism was significantly associated with life satisfaction. The health benefits of dietary fiber, vitamins, minerals, and phytochemicals from a plant-based diet have been well-studied, particularly in preventing metabolic syndrome, type 2 diabetes, coronary heart disease, and cancer [48,49,50]. Nevertheless, some studies have suggested that potential nutritional deficiencies among vegetarians in relation to vitamin B12, folates, and zinc might lead to adverse psychological outcomes [51]. A cross-sectional study of 1051 older adults in China reported that vegetarian diets were associated with a greater risk of depressive symptoms, particularly in older men [52]. However, another cross-sectional study of 219 Australian vegans and vegetarians showed that higher-quality dietary patterns were associated with a reduced risk of depressive symptoms [53]. In addition, a prospective cohort study following 12,062 Taiwanese people showed that vegetarians had a significantly lower risk of developing subsequent depressive disorders compared to non-vegetarians [54]. Regarding research on well-being, some studies have shown that vegetarians had poorer subjective well-being than omnivores [55,56], while others have reported that a vegan diet or increased consumption of fruits and vegetables was positively related to well-being [57,58]. Moreover, a study of 12,905 participants in Germany and 15,532 Australians concluded that the effect of diet on subjective well-being was either nonexistent or negligible [59]. In contrast, our study supported the notion that adopting a vegetarian diet could improve life satisfaction. A possible explanation is that vegetarianism aligns with the organizational culture of our volunteers. The Buddhist Compassion Relief Tzu Chi Foundation encourages their volunteers to consume a vegetarian diet for moral and environmental reasons. Feelings of belongingness have been shown to improve physiological well-being [60]. In addition to this plausible social–psychological reason, the role of specific nutrients in a plant-based diet with life satisfaction should be explored. 

This study also aimed to explore whether hedonic and eudaimonic motives could affect life satisfaction in our participants. Motives pertain to the personal reasons people have for their behavior and represent the priorities a person habitually pursues. When deciding whether to engage in an activity, individuals with hedonic motives seek pleasure, enjoyment, comfort, and absence of distress, whereas those with eudaimonic motives seek growth, meaning, authenticity, and excellence [29]. In our study, participants with eudaimonic motives, but not hedonic motives, were found to have significantly higher levels of life satisfaction. This result is not surprising because contributing one’s ability to society in an altruistic manner is intrinsic to volunteer work. A survey of 4085 Australian volunteers about their motivations showed that those with other-oriented motives had better well-being than those with self-oriented motives [61]. Our finding is also in line with that reported in a study on Chinese college students, which showed that eudaimonic motives in daily activities were positively associated with life satisfaction and meaning in life, while hedonic motives were not significantly associated with either indicator of well-being [62]. Furthermore, eudaimonic motives are generally encouraged in East Asian cultures, particularly among Buddhists. From a Buddhist standpoint, material gains and bodily pleasures are not true happiness, as these hedonic desires are transitory and illusory. Thus, the Eastern conceptualization of happiness aligns more with eudaimonic motive-based happiness [63]. The emphasis on the contribution to society or the community in Chinese culture was also supported by a bibliometric study of 162 publications from 2000 to 2021 comparing volunteer motivation between Chinese and American volunteers [64]. Therefore, future research should explore strategies to promote the eudaimonic motives of formal volunteers to achieve an optimal level of life satisfaction. For instance, a eudaimonic intervention based on counting blessings was shown to enhance self-reported gratitude, optimism, life satisfaction, and decreased negative affect in adolescents [65]. 

This study has some limitations that need to be considered. First, the cross-sectional design precluded the establishment of a causal direction of the relationships analyzed. The issue of endogeneity cannot be ruled out, as there is potential bidirectional relationship between self-esteem and life satisfaction. Nevertheless, results from a longitudinal study suggested that self-esteem tended to be an antecedent rather than a consequence of social support in adults [14]. Another study reported that self-esteem and behavioral activation, both independent and serially, mediated the positive association between cheerfulness and life satisfaction [64]. Hence, these studies suggested that individuals with higher levels of self-esteem tended to predict greater life satisfaction. Future research could utilize longitudinal designs to confirm the directionality of this relationship. Second, as in the case with any self-report instrument, respondents can consciously distort their response if they are motivated to do so. Therefore, the presence of social desirability bias could not be ruled out. Third, each respondent was offered a box of medical masks as an incentive to participate in the study. This incentive could have made participants feel compelled to respond positively to the survey questions. However, since the incentive was relatively small, the potential impact of response bias on the study results would be minimal. Fourth, all participants were formal volunteers recruited from the Buddhist Compassion Relief Tzu Chi Foundation. Whether the observed associations can be generalized to volunteers of other non-religious, non-governmental organizations will require further investigation. 

## 5. Conclusions

Findings from this study suggested a positive association between life satisfaction and self-esteem in older adults engaging in formal volunteering. In addition, adhering to a vegetarian diet, volunteering for five or more days per week, and possessing eudaimonic motives were also significantly associated with higher levels of life satisfaction. Future studies should explore strategies to increase self-esteem among older volunteers and thereby improve their life satisfaction levels.

## Figures and Tables

**Table 1 ijerph-20-04934-t001:** Demographic information (N = 186).

Variable	N	%
Sex		
male	40	21.5
female	146	78.5
Age group, years		
65–69	71	38.2
70–74	65	34.9
>75	50	26.9
Educational level		
primary school and below	79	42.5
junior and senior high school	73	39.2
university and above	34	18.3
Marital status		
being married	120	64.5
other	66	35.5
Employment status		
not employed	163	87.6
full-time or part-time employed	23	12.4
Dietary habits		
vegetarian (strict or lacto-ovo)	126	67.7
flexitarian	60	32.3
Chronic disease		
no	100	53.8
yes	86	46.2
Sleep duration, hours		
<7	120	64.5
≥7	66	35.5
Duration of volunteering, years		
0–9	26	14.0
10–20	75	40.3
>20	85	45.7
Frequency of volunteering, days/week		
<5	144	77.4
≥5	42	22.6
Main type of volunteering		
environmental protection	87	46.8
charity	30	16.1
community	24	12.9
medical	18	9.7
educational	11	5.9
humanism	8	4.3
blood marrow registry	8	4.3

SD, standard deviation.

**Table 2 ijerph-20-04934-t002:** Descriptive statistics for the scores of the Rosenberg Self-Esteem Scale, Hedonic and Eudaimonic Motives for Activities-Revised scale, and Satisfaction with Life Scale.

Scale	Mean (SD)	Median (min, max)
Rosenberg Self-Esteem Scale	29.5 (3.9)	29 (20, 40)
HEMA-R		
Hedonic subscale	23.3 (4.5)	24 (10, 34)
Eudaimonic subscale	27.9 (3.8)	25.5 (17, 35)
Satisfaction with Life Scale	26.8 (4.3)	28 (12, 35)

HEMA-R, Hedonic and Eudaimonic Motives for Activities-Revised; max, maximum; min, minimum; SD, standard deviation.

**Table 3 ijerph-20-04934-t003:** Hierarchical linear stepwise regression analysis of the Satisfaction with Life Scale score (N = 186).

Variable	Model 1			Model 2			Model 3			Model 4		
	β	Std. β	*p*	β	Std. β	*p*	β	Std. β	*p*	β	Std. β	*p*
Intercept	25.18	–	<0.001	24.78	–	<0.001	12.28	–	<0.001	7.93	–	<0.001
Vegetarian	2.40	0.261	<0.001	2.26	0.246	<0.001	1.29	0.140	0.033	1.31	0.143	0.027
Frequency of volunteering, ≥5 vs. <5 days/week				2.20	0.214	0.002	1.88	0.184	0.004	1.65	0.161	0.011
HEMA-R, eudaimonic subscale score							0.47	0.416	<0.001	0.40	0.353	<0.001
Rosenberg Self-Esteem Scale score										0.22	0.199	0.003
Durbin–Watson statistic	1.96			1.98			2.08			2.03		
Adjusted R^2^	0.063			0.104			0.263			0.294		
ΔR^2^	–			0.046			0.161			0.035		
*p*	–			0.002			<0.001			0.003		

β, unstandardized regression coefficient; ∆R^2^, the change in R^2^ values from one model to another; R^2^, the proportion of explained variance in the Satisfaction with Life Scale score by the model; std. β, standardized regression coefficient.

## Data Availability

The datasets generated for this study are available upon request from the corresponding author.

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
