# Peer review of "Life Satisfaction and Self-Esteem in Older Adults Engaging in Formal Volunteering: A Cross-Sectional Study in Taiwan"

_ijerph, 2023, doi:10.3390/ijerph20064934_

Round 1
Reviewer 1 Report
This study has been well-designed and performed. The results and discussion are well written. The only thing I request from the authors is a STROBE form checklist, which must be mentioned in the "Materials and Mathods" section. The fillable form can be downloaded from https://www.strobe-statement.org/checklists/.
Author Response
Reviewer 1’s Comment:
This study has been well-designed and performed. The results and discussion are well written. The only thing I request from the authors is a STROBE form checklist, which must be mentioned in the "Materials and Mathods" section. The fillable form can be downloaded from https://www.strobe-statement.org/checklists/.
Response from the authors: We appreciate the reviewer’s suggestion and have included a STROBE checklist as an appendix.
Reviewer 2 Report
This study examines the relationship between life satisfaction and self-esteem in older adults engaging in formal volunteering in Taiwan. The topic is very interesting. But the manuscript needs more revisions before accepted.
1. Although the authors review the relevant literature on life satisfaction, they should pay more attentions on the influence of self-esteem on life satisfaction. A detailed theoretical framework is needed here.
2. The authors tested the model between life satisfaction and self-esteem. But there are no enough discussions or relative literature on the causal relationships between these two variables. More discussions are needed.
3. The sample size is too small to test the regression models. In addition, the authors should take endogeneity issue into account.
4. The authors should provide with more information on the sampling method and more sample characteristics.
Author Response
Reviewer 2, Comment #1: Although the authors review the relevant literature on life satisfaction, they should pay more attentions on the influence of self-esteem on life satisfaction. A detailed theoretical framework is needed here.
Response to Reviewer 2, Comment #1: We greatly appreciate the reviewer for this suggestion. We have added a new paragraph in the Introduction to describe the Self-Evaluation Maintenance model as a theoretical framework (page 2, line 55 to 64)
Reviewer 2, Comment #2: The authors tested the model between life satisfaction and self-esteem. But there are no enough discussions or relative literature on the causal relationships between these two variables. More discussions are needed.
Response to Reviewer 2, Comment #2: We appreciated the reviewer’s suggestion and have added a paragraph in the Discussion section (page 6, line 229 to page 7, line 238).
Reviewer 2, Comment #3: The sample size is too small to test the regression models. In addition, the authors should take endogeneity issue into account.
Response to Reviewer 2, Comment #3: We based our sample size on a small-medium effect size with eight predictors in a linear multiple regression, a 0.05, and a power of 80%.
Regarding the issue of endogeneity, we agree that endogeneity may occur when there is a bidirectional relationship between the explanatory variables and the outcome variable, which is possible in our study. However, one prospective study suggested that self-esteem tended to be an antecedent rather than a consequence of social support in adulthood (reference no. 15). Another study by Lau et al. (reference 66) suggested that self-esteem and behavioral activation, both independent and serially, mediated the positive association between cheerfulness and life satisfaction. Thus, these studies suggested that individuals with higher levels of self-esteem tend to predict greater life satisfaction. Nevertheless, we have added the potential issue of endogeneity in the limitation section. [page 8, line 304–311].
Reviewer 2, Comment #4: The authors should provide with more information on the sampling method and more sample characteristics.
Response to Reviewer 2, Comment #4: We have included additional information about the sampling method in the Methods section of the manuscript. Convenience sampling was used in this study. A trained interviewer went to the chapter office two to three times a week over the period November 1 to 30, 2022 to conduct the survey. A total of 191 questionnaires were distributed, but five contained missing responses, resulting in 186 questionnaire for analysis. (page 2, line 91–96). In addition, basic characteristics of the participants are included in Table 1.
Reviewer 3 Report
The purpose of this study was to explore the association between life satisfaction and self-esteem among older people engaged in formal volunteering in a non-governmental organisation in Taiwan. This is a timely and generally well-written article. A reading of this article raises one minor point reported below.
1/In the section on "study limitations" you should discuss the potential impact of the fact that each respondent was offered a box of medical masks worth 100 new Taiwanese dollars (about 3 US dollars) for participating in the study.
Author Response
We greatly appreciate the reviewer’s suggestion and have added the following to the limitation section:
Each respondent was offered a box of medical masks as an incentive to participate in the study. This incentive could have made participants feel compelled to respond positively to the survey questions. However, since the incentive was relatively small, we believe that the potential impact of response bias on the study results would be minimal.